# Structure of the prefusion-locking broadly neutralizing antibody RVC20 bound to the rabies virus glycoprotein

Jan Hellert [1], Julian Buchrieser [2,8], Florence Larrous [3,8], Andrea Minola[4,8], Guilherme Dias de Melo [3,8], Leah Soriaga[5], Patrick England[6], Ahmed Haouz [7], Amalio Telenti[5], Olivier Schwartz[2], Davide Corti [4], Hervé Bourhy [3]* & Félix A. Rey [1]*

Rabies virus (RABV) causes fatal encephalitis in more than 59,000 people yearly. Upon the bite of an infected animal, the development of clinical disease can be prevented with post-exposure prophylaxis (PEP), which includes the administration of Rabies immunoglobulin (RIG). However, the high cost and limited availability of serum-derived RIG severely hamper its wide use in resource-limited countries. A safe low-cost alternative is provided by using broadly neutralizing monoclonal antibodies (bnAbs). Here we report the X-ray structure of one of the most potent and most broadly reactive human bnAbs, RVC20, in complex with its target domain III of the RABV glycoprotein (G). The structure reveals that the RVC20 binding determinants reside in a highly conserved surface of G, rationalizing its broad reactivity. We further show that RVC20 blocks the acid-induced conformational change required for membrane fusion. Our results may guide the future development of direct antiviral small molecules for Rabies treatment.

[1] Structural Virology Unit, Institut Pasteur, CNRS UMR 3569, 25-28 rue du Docteur Roux, Cedex 15, 75724 Paris, France. [2] Virus and Immunity Unit, Institut Pasteur, CNRS UMR 3569, 25-28 rue du Docteur Roux, Cedex 15, 75724 Paris, France. [3] Lyssavirus Epidemiology and Neuropathology Unit, Institut Pasteur, 25-28 rue du Docteur Roux, Cedex 15, 75724 Paris, France. [4] Humabs BioMed SA, a subsidiary of Vir Biotechnology Inc., Via dei Gaggini 3, 6500 Bellinzona, Switzerland. [5] Vir Biotechnology Inc, San Francisco, CA 94158, USA. [6] Molecular Biophysics Platform C2RT, Institut Pasteur, CNRS UMR 3528, 25-28 rue du Docteur Roux, Cedex 15, 75724 Paris, France. [7] Crystallography Platform C2RT, Institut Pasteur, CNRS UMR 3528, 25-28 rue du Dr. Roux, Cedex 15, 75724 Paris, France. [8] These authors contributed equally: Julian Buchrieser, Florence Larrous, Andrea Minola, Guilherme Dias de Melo. *email: herve.bourhy@pasteur.fr; felix.rey@pasteur.fr

Rabies virus (RABV) belongs to phylogroup I of the *Lyssavirus* genus within the *Rhabdoviridae* family of the *Mononegavirales* order[1]. It is a zoonotic virus found almost ubiquitously worldwide in different animal reservoirs, including domestic and wild canids and bats. Despite significant efforts, most countries face severe difficulties with RABV control[2,3], and in fact the virus has been eliminated only from a few developed countries by mass vaccination of wild and domestic canines[4]. Today, an estimated 3 billion people are living at risk of contracting rabies through the bite of infected animals, mainly in Asia and Africa, where half of the victims are children under the age of 15 (refs. [5,6]). Still, 19–50 million people receive postexposure prophylaxis (PEP) each year[4]. Moreover, rabies disease with equally fatal outcome can also be caused by a number of non-RABV lyssaviruses, many of which use bats as their primary vector.

Following the bite of a potentially infected animal, administration of three doses of vaccine over the first week and one dose of Rabies immunoglobulin (RIG) without delay is recommended in order to eliminate the virus before it enters the nervous system[7,8]. Recombinant antibody preparations are preferred over traditional serum-derived polyclonal human or equine RIG, as they can be produced in large scale with minimal batch-to-batch variation ensuring improved safety. Yet, the only monoclonal antibody licensed to date does not provide full coverage against all circulating RABV strains, thus posing a risk for lack of efficacy and viral escape[9] (Rabishield by Mass Biologics and Serum Institute of India Pvt. Ltd.). One of the best broadly neutralizing monoclonal antibodies (bnAbs) currently known, RVC20, was shown to not only exhibit a higher neutralization potency against 100% of 35 tested RABV strains from across the world, but also to neutralize a wider range of non-RABV lyssaviruses[9]. Moreover, RVC20 protected hamsters from lethal RABV infection in combination with another bnAb, RVC58, which targets a distinct antigenic site[9]. The sole target of all neutralizing antibodies is RABV G, but despite its medical relevance, no structural data are available for this envelope protein yet. In order to understand the molecular determinants for broad and efficient RABV neutralization, we here set out to determine the X-ray structure of RVC20 in complex with its antigen.

## Results

**X-ray structure of the complex.** The ectodomain of the rhabdovirus G protein is divided into four distinct subdomains denoted I, II, III and IV (Fig. 1a), as first observed in the structure of vesicular stomatitis virus (VSV) G[10,11]—a member of the *Vesiculovirus* genus in the *Rhabdoviridae* family. The G domain nomenclature is not to be confused with the RABV antigenic site designation introduced in earlier literature[12,13]. RVC20 recognizes antigenic site I on RABV G domain III, which is folded as a Pleckstrin homology (PH) domain and is the most exposed domain of the rhabdovirus prefusion spike, making it a dominant target for the adaptive humoral immune response[9,11]. Based on its homology with VSV G (Supplementary Fig. 1), we generated a recombinant domain III construct encompassing RABV G residues E31-V56 and N182-D262 (Fig. 1a). We determined its crystal structure in complex with the single-chain variable fragment (scFv) of RVC20 to a resolution beyond 2.7 Å and refined the atomic model to a final $R_{free}$ value of 0.22 (Fig. 1b, Table 1).

The buried surface area (BSA) on the antigen spans 720 Å²; a typical value for IgG immune complexes[14]. The paratope is composed of the complementarity-determining region (CDR) L3 of the light chain and all three CDRs of the heavy chain, with the two antibody chains respectively contributing 20% and 80% of the total BSA (Fig. 1c, d). The epitope is composed of three short polypeptide segments of domain III, two of which are linked through a disulfide bond (C189–C228, conserved in G from viruses of 13 out of the 20 *Rhabodoviridae* genera) that is tightly embraced by the CDR H3 (Fig. 1c, d). Other epitope residues with large contributions to the BSA are also well conserved across RABV strains (Fig. 1c, Table 2). Analysis of 1412 unique full-length RABV G amino acid sequences from the database indicated that positions 190, 194 and 231 display the highest variability across strains, with L231 making only minor contributions to the BSA (Fig. 1c). The most frequent polymorphisms at these positions (i.e. N194S, N194T, L231P and L231S) had no effect on RVC20 neutralization in the context of natural strains[9]. In addition to side chain conservation, the main chain is likely an important binding determinant. In particular, the β-turn at residues C228 and G229, recognized by RVC20 through direct hydrogen bonds (Fig. 1d), is predicted to be identical in G from all lyssaviruses.

**Epitope mutagenesis.** In support of these observations, we found that RVC20 neutralization is relatively robust to single point mutations in the epitope. We selected residues for mutation based on their predicted impact on epitope topology (Fig. 1e, Supplementary Fig. 2). For example, the D190 side chain makes a π-stacking interaction with $Y52_{VH}$ and hydrogen bonds with $S54_{VH}$ and $S56_{VH}$ (Fig. 1d); yet introduction of point mutation D190S by reverse genetics hardly conferred any resistance to RVC20 (Fig. 1e). Similarly, the V230 side chain makes hydrophobic contacts with $Y58_{VH}$ and $Y94_{VK}$ (Fig. 1d); yet the variant V230M, as found in Irkut virus of phylogroup I, or V230K as found in most phylogroup II or III/IV lyssaviruses (Fig. 1c), only mildly reduced neutralization by RVC20 (Fig. 1e). We observed the most pronounced but still mild gain in resistance to RVC20 with point mutation K226T, as found in several phylogroup II or III/IV viruses (Fig. 1c, e). The K226 side chain makes contacts with CDR H3, forms a hydrogen bond with $T93_{VK}$, and participates in the only salt bridge throughout the interface with $D92_{VK}$, explaining the phenotype of the mutant (Fig. 1d). The above single point mutations have similarly mild effects on mAb binding as on neutralization (Supplementary Fig. 3). These findings highlight the overall high conservation of the epitope, with only a few natural variations in distantly related lyssaviruses such as Lagos bat virus (LBV), a phylogroup II lyssavirus, compromising RVC20 efficacy. Indeed, the G protein of LBV, which was shown to resist RVC20 neutralization[9], displays 226T and 230K at the epitope. Using a lentivirus pseudotype system, we converted these two residues of LBV G to 226K and 230V and found that RVC20 neutralization was restored, further confirming the importance of these two side chains for antibody–antigen recognition (Fig. 1f, Supplementary Fig. 2).

**Affinity maturation of RVC20.** RVC20 shares 94% sequence identity in its variable domains with the inferred unmutated ancestor (UA) (Fig. 2a). Somatic hypermutation (SHM) has introduced at least 14 amino acid changes, only two of which, $N92D_{VK}$ and $S93T_{VK}$, map directly to the paratope (Fig. 2b). As discussed above, these two residues interact with K226 of the antigen via a salt bridge and two hydrogen bonds. We found that only the $D92N_{VK}$ reversion, which breaks the salt bridge, but not the $T93S_{VK}$ reversion, which leaves the interactions unchanged, had a mild negative effect on neutralization (Fig. 2c). Likewise, the $S98Y_{VH}$ reversion, positioned within CDR H3 but not in direct contact with the antigen, did not affect neutralization (Fig. 2c). Nevertheless, the UA neutralized RABV 20-fold less efficiently than the mature RVC20, a difference that cannot be attributed to the SHM $N92D_{VK}$ at the paratope alone (Fig. 2c). It

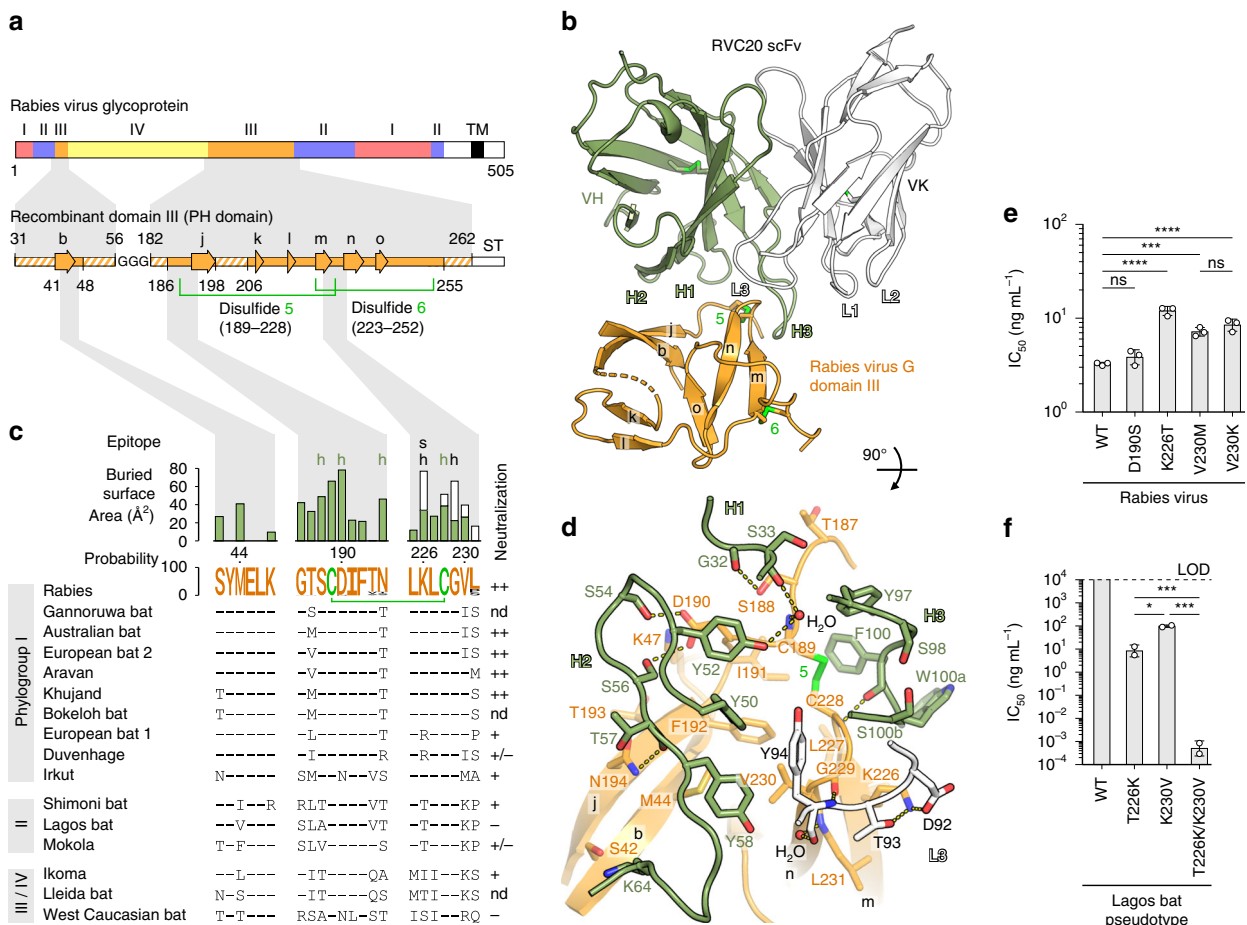

**Fig. 1 RABV G recognition by RVC20. a** Domain organization of RABV G (top row) as inferred by homology to VSV G[10], and design of the recombinant domain III construct for structure determination (bottom row). Hatched fields in the construct denote unresolved regions in the X-ray structure. β strands are shown as arrows labeled in lower case in accordance with the VSV G structure[10]. TM transmembrane region, ST Strep tag. **b** Crystal structure of the complex between RABV G domain III and the RVC20 scFv. The variable domain of the heavy chain (VH) is shown in green, the variable domain of the kappa light chain (VK) is shown in white and the antigen is shown in orange. The CDRs of the mAb and the β strands and disulfide bonds of the antigen are labeled. **c** Sequence conservation of the tripartite epitope. The RABV G sequence is displayed as a sequence logo, indicating the conservation per residue across 1412 unique full-length RABV G sequences in GenBank (details are listed in Table 2). The color-coded bar chart shows the BSA per RABV G residue in the complex contributed by heavy chain (green) and light chain (white) residues. h, hydrogen bond involved; s, salt bridge involved. Differences in epitope sequence across the *Lyssavirus* genus are shown below, with the corresponding neutralizing potency of RVC20 qualitatively summarized to the right[9]: ++, strong; +, attenuated; −, not detected; +/−, isolate-dependent; nd, not determined. **d** Detail of the interaction interface. Residues on both sides of the interface are labeled and are shown as sticks with oxygen atoms in red and nitrogen atoms in blue. Secondary-structure elements and disulfide bonds are labeled as in **b**. **e** Neutralization of recombinant RABV mutants with mAb RVC20 on BSR cells 48 h after infection; $n = 3$ independent experiments. **f** Neutralization of wild-type and mutant Lagos bat virus G-pseudotyped lentiviruses with mAb RVC20 on BHK-21 cells 72 h after infection. LOD, limit of detection; $n = 2$ independent experiments. Data are displayed as means ± s.d. Statistical analysis was performed using Tukey's test with $\alpha = 0.05$. ****$P < 0.0001$; ***$P < 0.001$; *$P < 0.05$; ns, not significant ($P > 0.05$). Source data are provided as a Source Data file.

is thus instructive that four SHMs near the VH/VK interface, Y35S$_{VH}$, S35bN$_{VH}$, F100fL$_{VH}$ and Y87F$_{VK}$, together contributed with a sixfold increase in neutralization efficiency relative to the UA. These changes therefore likely affect the paratope indirectly through improved relative orientations of the two chains (Fig. 2b, c). Indeed, we observed that the SHMs on both the heavy and light chains contributed similarly to affinity maturation (Fig. 2c). The neutralization capacity of all tested RVC20 variants correlated with their affinity (Supplementary Fig. 3). Overall, the small number of SHMs and the intrinsic neutralization activity of the inferred UA suggest that RVC20-like antibodies utilizing the same germline gene segments may frequently be selected in response to vaccination or infection.

**Neutralization mechanism.** Rhabdovirus G is a class III membrane fusion protein that induces fusion of the viral envelope with endosomal membranes in order to deliver the viral genome into the cytoplasm[15]. Fusion requires G to undergo a specific conformational change upon exposure to the acidic environment of the endosome. This rearrangement is known to be reversible in vitro[16], and the structures of both the alkaline-pH prefusion conformation and the acidic-pH postfusion conformation have been described for the VSV G ectodomain[10,11]. Comparison of our immune complex to these previously determined structures suggested that the RVC20 epitope should be accessible only in the alkaline-pH prefusion conformation of RABV G (Fig. 3a, b). We experimentally confirmed that association of the purified RABV G ectodomain with RVC20 slows down as the pH drops from 8.0 to 7.0 and then to 5.5, where the low-pH conformation predominates (Fig. 3c). Yet, dissociation remains slow and is largely unaffected by pH (Fig. 3c). As suggested by the comparison to VSV G, the pH-dependency of the association step appeared to be

## Table 1 Crystallographic data collection and refinement statistics.

**Data collection and processing**

| | |
|---|---|
| Space group | $P\,4_1\,2_1\,2$ |
| Cell dimensions | |
| $a, b, c$ (Å) | 81.95, 81.95, 155.93 |
| $\alpha, \beta, \gamma$ (°) | 90, 90, 90 |
| Resolution range[a] (Å) | 46.51−2.59 (2.72−2.59) |
| Ellipsoidal highest resolution[b] (Å) / direction | 2.58 / $a\star$ |
| | 2.58 / $b\star$ |
| | 2.76 / $c\star$ |
| Number of unique reflections[a] | 15,457 (774) |
| $R_{sym}$[a] | 0.10 (3.69) |
| $R_{pim}$[a] | 0.02 (0.70) |
| $<I/\sigma(I)>$ [a] | 21.4 (1.1) |
| $CC_{1/2}$[a] | 1.00 (0.66) |
| Completeness, spherical[a] (%) | 89.5 (33.3) |
| Completeness, ellipsoidal[a,b] (%) | 94.9 (55.5) |
| Redundancy[a] | 25.7 (28.1) |
| **Structure refinement** | |
| Resolution range[a] (Å) | 43.89−2.59 (2.68−2.59) |
| Number of unique reflections[a] | 15,453 (321) |
| $R_{work}$ / $R_{free}$[a] | 0.19 / 0.22 (0.36 / 0.39) |
| Number of atoms | |
| Protein | 2323 |
| Ligands/ions | 4 |
| Water | 303 |
| Average $B$-factor (Å$^2$) | |
| Protein | 106 |
| Ligands/ions | 124 |
| Water | 93 |
| R.m.s. deviations | |
| Bond lengths (Å) | 0.003 |
| Bond angles (°) | 0.62 |

[a]Values in parentheses are for the highest-resolution shell.
[b]The datasets were anisotropically truncated using the STARANISO web server. An ellipsoid was fitted to the anisotropic cut-off surface to provide approximate resolution limits along three directions in reciprocal space. The real cut-off surface is only approximately ellipsoidal and the directions of the worst and best resolution limits may not correspond with the reciprocal axes.

caused by epitope masking via domain rearrangements, since the isolated domain III bound RVC20 in a pH-independent manner (Fig. 3c). The reduced association rate at low pH is thus consistent with an equilibrium shift of G toward its acidic-pH postfusion form. We further confirmed this effect using HEK 293T cells expressing full-length RABV G on their surface, where RCV20 associated faster with the cells at pH 7.0 than at pH 5.5, while dissociation remained unaffected by pH (Fig. 3d). In contrast, a G-specific poorly neutralizing control mAb, RVC68, for which the epitope is unknown, bound RABV G-expressing HEK 293T cells preferentially at acidic pH (Fig. 3d).

The high stability of the immune complex even at pH 5.5 suggests that RVC20 efficiently locks G in its prefusion state or in an early intermediate conformation, preventing the structural rearrangements that drive membrane fusion. Indeed, we showed that RVC20 completely inhibited G-mediated syncytia formation at a concentration of 800 ng mL$^{-1}$ in a cell–cell fusion assay (Fig. 3e). The high degree of conservation of the RVC20 epitope across RABV strains and related lyssaviruses is probably linked to its involvement in the interaction between domains III and IV during fusion (Fig. 3a, b), which likely limits the mutation rate at this interface. Targeting this epitope and thereby blocking membrane fusion, which is an essential and universal step of the viral life cycle, is a safer approach than the inhibition of cell-type-specific receptor binding, which may protect only a subset of target cells from infection[1]. Our study has thus characterized a vulnerable site on *Lyssavirus* G, which could be targeted not only for mAb-based prophylaxis, but also for future therapeutic applications in cases where the virus has already entered the nervous system. High-throughput screening of small molecules competing with RVC20 or a structure-based design of short peptide mimics of the bnAb's CDRs could be employed in a first step of drug development, as recently demonstrated for the influenza virus fusion protein[17,18]. Our results therefore also provide the groundwork for the design of low-molecular-weight fusion inhibitors capable of crossing the blood–brain barrier to extend the narrow time window for countermeasures against lyssavirus infection beyond what is currently feasible with PEP.

## Table 2 Percent conservation and frequency of amino acids per position of the RVC20 epitope across 1412 unique full-length RABV G sequences in GenBank.

| Position | Consensus | BSA[a] (Å$^2$) | Conservation (%) | Observed amino acid counts |
|---|---|---|---|---|
| 42 | S | 27 | 99.9 | S-1411, P-1 |
| 44 | M | 41 | 99.1 | M-1399, I-7, L-4, V-2 |
| 47 | K | 10 | 99.9 | K-1410, R-2 |
| 186 | G | 43 | 99.1 | G-1399, R-10, E-3 |
| 187 | T | 33 | 99.3 | T-1402, M-8, K-1, A-1 |
| 188 | S | 50 | 99.4 | S-1404, P-5, Y-2, F-1 |
| 189 | C | 67 | 99.9 | C-1411, R-1 |
| 190 | D | 79 | 98.1 | D-1385, N-27 |
| 191 | I | 23 | 99.9 | I-1411, T-1 |
| 192 | F | 22 | 99.7 | F-1408, L-3, S-1 |
| 194 | N | 47 | 87.8 | N-1240, T-77, S-61, Y-33, K-1 |
| 225 | L | 12 | 99.1 | L-1399, M-10, V-2, I-1 |
| 226 | K | 74 | 99.6 | K-1406, M-3, R-3 |
| 227 | L | 28 | 100.0 | L-1412 |
| 228 | C | 49 | 99.9 | C-1411, S-1 |
| 229 | G | 67 | 100.0 | G-1412 |
| 230 | V | 38 | 99.8 | V-1409, I-3 |
| 231 | L | 16 | 76.4 | L-1078, P-192, S-140, H-1, T-1 |

[a]Buried surface area, as determined by the PDBePISA web server[33].

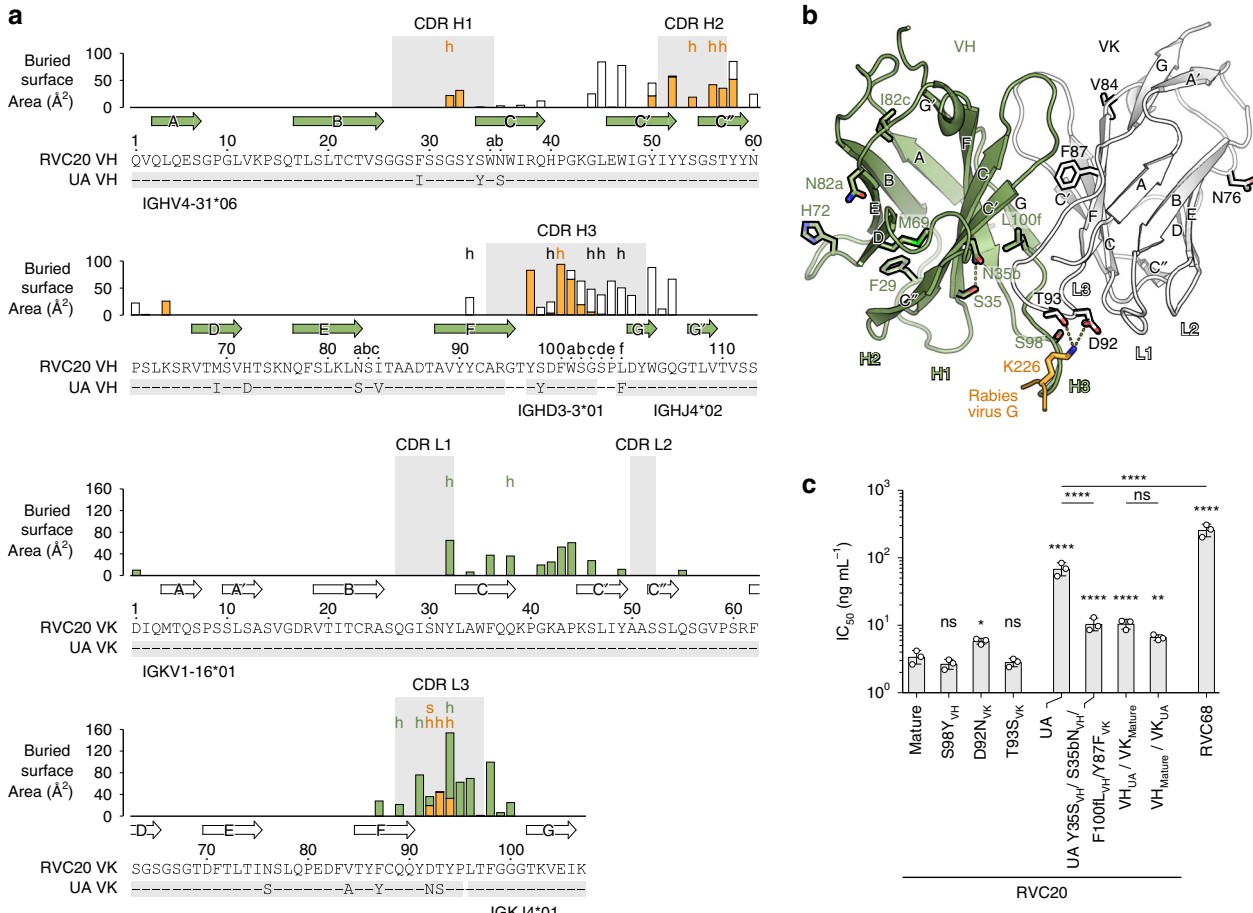

**Fig. 2 RVC20 sequence analysis. a** Annotation of the RVC20 variable heavy (VH) and variable kappa light (VK) sequences using the Kabat numbering[32]. Differences to the most closely matching germline sequences as determined by IGMT/V-QUEST[27] are indicated below each sequence, defining the inferred unmutated ancestor (UA) sequences. Arrows denote β strands. The color-coded bar chart indicates the BSA per residue in contacts with the light chain, the heavy chain, or the antigen in white, green or orange, respectively; h, hydrogen bond involved; s, salt bridge involved. The CDRs (IMGT convention) are highlighted in gray background. **b** Positions of the inferred SHMs on the structure of the RVC20 scFv. RABV G residue K226 interacting with D92$_{VK}$ and T93$_{VK}$ is shown in orange. **c** Neutralization of RABV with RVC20 variants or with poorly neutralizing RVC68 on BSR cells 48 h after infection; $n = 3$ independent experiments. Data are displayed as means ± s.d. Statistical analysis was performed using Tukey's test with $\alpha = 0.05$. ****$P < 0.0001$; ***$P < 0.001$; **$P < 0.01$; *$P < 0.05$; ns, not significant ($P > 0.05$). Source data are provided as a Source Data file.

## Methods

**Mammalian cell culture**. HEK293-T clone 17 cells (ATCC CRL-11268), BHK-21 clone 13 cells (ATCC CCL-10) and BSR cells (a BHK-21 clone, kindly provided by Monique Lafon, Institut Pasteur, Paris) were cultured at 37 °C, 5% CO$_2$, in Dulbecco's minimal essential medium (DMEM) supplemented with 10% fetal calf serum (FCS). The BSR-T7 cells[19] (kindly provided by Karl-Klaus Conzelmann, Max von Pettenkofer Institute and Gene Center, Munich) for reverse genetics were cultured in Glasgow medium supplemented with 10% FCS, 2% (0.59 g L$^{-1}$) tryptose phosphate, 1% non-essential amino acids and 0.1% (50 μg mL$^{-1}$) geneticin.

**Recombinant protein preparation for crystallization**. Domain III of RABV G (strain 9147FRA, GenBank: AF401286) and the RVC20 scFv were produced in Drosophila S2 cells (Gibco) expressing codon-optimized synthetic genes (Invitrogen) with C-terminal Strep tags (sequence: GGWSHPQFEK) downstream of a BiP secretion signal (sequence: MKLCILLAVVAFVGLSLG) within the pMT expression vector. Domain III was constructed joining codons for E31-V56 and N182-D262 with a short linker of three glycine codons. The RVC20 scFv was constructed joining the VH and VK coding regions with a glycine-serine linker of 20 codons (sequence: GGGGS GGGGS GGGGS GGGGG). All cloning primers are listed in Supplementary Table 1.

S2 cells were grown at 28 °C in HyClone SFM4Insect medium with L-glutamine (GE Healthcare) supplemented with 25 U mL$^{-1}$ penicillin/streptomycin (Gibco). Expression plasmids were co-transfected with the selection plasmid pCoPURO[20] at a mass ratio of 20:1 using Effectene transfection reagent (Qiagen) according to the manufacturer's instructions. Polyclonal stable S2 cell lines were established by selection with 7.5 μg mL$^{-1}$ puromycin (Invivogen), which was added to the medium 40 h after transfection. Cultures were expanded to 1 L of 10$^7$ cells mL$^{-1}$ in Erlenmeyer flasks shaking at 100 rpm and at 28 °C. Recombinant protein

expression was then induced with 500 μM CuSO$_4$. Cell supernatants were harvested 1 week after induction, concentrated to 50 mL on a 10-kDa MWCO PES membrane (Sartorius), pH-adjusted with 0.1 M Tris-Cl pH 8.0, cleared from biotin with 15 μg mL$^{-1}$ avidin, cleared from precipitate by centrifugation at 4000×g for 15 min at 8 °C, and were then used for affinity purification on a 5-mL Strep-Tactin Superflow hc column (iba Life Science). The two proteins were further purified by gel permeation chromatography on a HiLoad Superdex 200 pg column (GE Healthcare) in 20 mM Tris-Cl pH 8.0, 150 mM NaCl, and were subsequently mixed at equal molar ratio. The complex was purified by gel permeation chromatography. The final sample was concentrated to a protein concentration of 18 mg mL$^{-1}$ in a 10-kDa MWCO PES Vivaspin centrifugal concentrator (Sartorius).

**Crystallization of the RABV G domain III with the RVC20 scFv**. Optimal crystals were obtained by the sitting-drop vapor diffusion method. A total of 0.7 μL of 18 mg mL$^{-1}$ complex in 20 mM Tris-Cl pH 8.0, 150 mM NaCl were added to 0.7 μL of reservoir solution containing 100 mM Tris-Cl pH 8.0, 300 mM CaCl$_2$, 22% w/v PEG4000. The drops were equilibrated against reservoir solution for 2 weeks at 18 °C. Crystals were then cryo-protected in 80 mM Tris-Cl pH 8.0, 240 mM CaCl$_2$, 17.6% w/v PEG4000, 20% v/v glycerol prior to conservation in liquid nitrogen.

**Crystallographic data collection and structure determination**. X-ray diffraction data were recorded on synchrotron beamline PX2 at SOLEIL in St Aubin, France, with an EIGER X 9M detector (Table 1). The wavelength was set to 0.9801 Å. Data were processed using XDS[21] and the STARANISO web server (Global Phasing Ltd.). Initial phases were obtained by molecular replacement in Phenix.MR[22] using a model of the RVC20 scFv generated with the Phyre2 web server for 3D homology

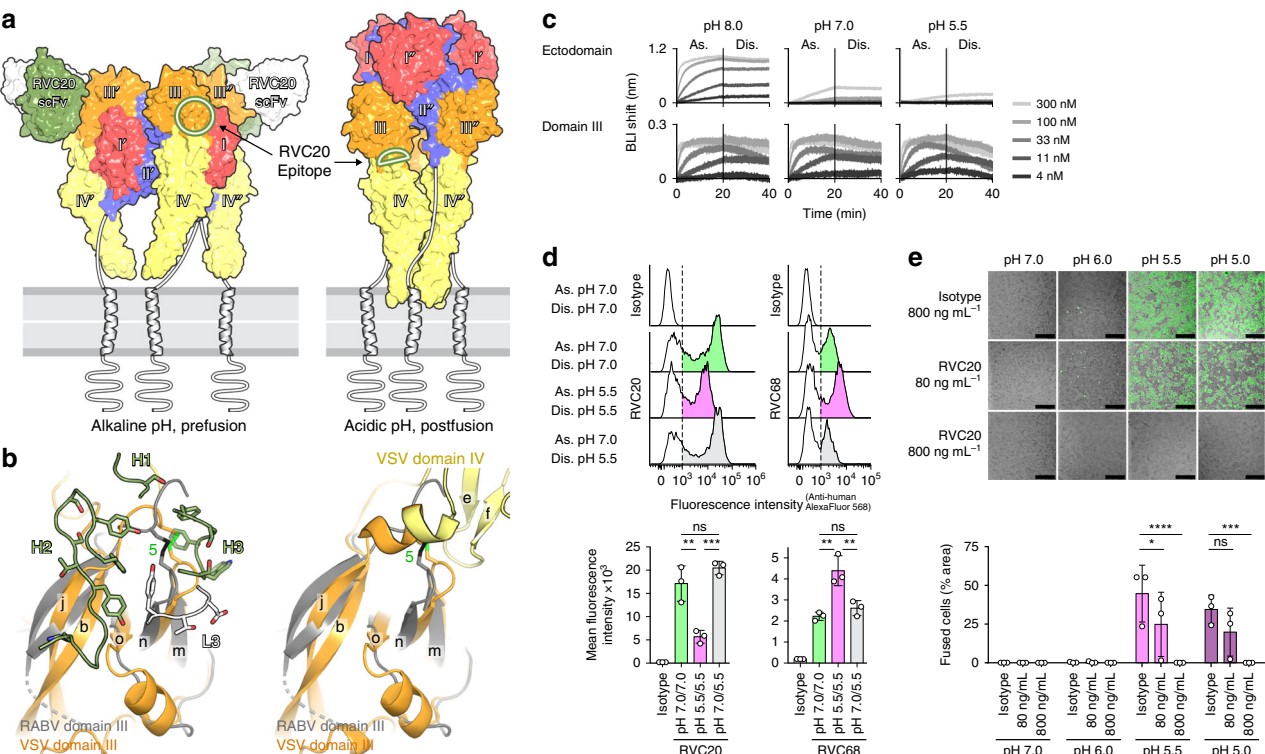

**Fig. 3 RVC20 locks RABV G in its prefusion state. a** Structure of the trimeric VSV G in its alkaline-pH prefusion[11] (left) and the acidic-pH postfusion conformation[10] (right) color-coded according to domains, as labeled. The RVC20 scFv/RABV G domain III complex was superposed onto VSV G domain III, and only the RVC20 moiety is shown. For clarity, the scFv is not shown on the front protomer, in which a circle marks the location of its epitope. The right panel shows that the epitope should become occluded by domain IV after the acidic-pH-triggered conformational change of G. **b** Detail of the VSV G region corresponding to the epitope (orange) superposed onto RABV G domain III (gray). The CDRs making the RVC20 paratope (green and white) are shown on the left panel. The right panel shows that the epitope is not accessible at acid pH, as domain IV buries a substantial part of the epitope area. The orientation is as in Fig. 1d. **c** Association (As.) and dissociation (Dis.) of the recombinant ectodomain (aa 1–403) or domain III alone (aa 31–56/182–262) to immobilized RVC20 at different pH values as determined qualitatively by biolayer interferometry (BLI). **d** Binding of RVC20 (left) or poorly neutralizing RVC68 (right) to RABV G-expressing HEK 293T cells in suspension. Association (As.) and washing (Dis.) was performed at the indicated pH values, and binding was assessed by flow cytometry in comparison to an isotype control mAb (top). The mean fluorescence intensity of mAb-bound cells (shaded area in histograms) is shown for $n = 3$ independent experiments (bottom). **e** Fusion inhibition by RVC20 in a GFP-split cell–cell fusion assay. Fusion of RABV G-expressing HEK 293T cells was determined upon exposure to the indicated pH in the presence of a nonspecific isotype control mAb or RVC20 at concentrations of 80 or 800 ng mL$^{-1}$; $n = 3$ independent experiments. Data are displayed as means ± s.d. Statistical analysis was performed using Tukey's test with $\alpha = 0.05$. ****$P < 0.0001$; ***$P < 0.001$; **$P < 0.01$; *$P < 0.05$; ns not significant ($P > 0.05$). Scale bars: 200 μm. Source data are provided as a Source Data file.

modeling[23]. Refinement was performed in iterations of manual model building in Coot[24] and automatic refinement in Phenix.Refine[22]. Ramachandran analysis of the final structure containing a single complex in the asymmetric unit indicates that 97% of the residues are in a favored conformation and 3% are in an allowed conformation. Phenix[22] further reports a Wilson B factor of 84.61 Å$^2$, 2.3% rotamer outliers and a clash score of 4.59. A visual assessment of the quality of the structure is provided with Supplementary Fig. 4.

**Epitope conservation analysis across RABV isolates.** RABV G sequences were downloaded from NCBI (2019/04/10) by taxon classification (taxon:11292)[25]. Sequences with "cell culture" in the "host" field of the GenBank record ($n = 129$) were excluded. Full-length G sequences with country source and collection date information ($n = 2875$) were collapsed to a non-redundant set of 1412 sequences for analysis.

To determine amino acid identity and coverage at epitope residues, each putative G sequence was aligned pairwise against the reference G sequence of the CVS-11 isolate (GenBank: ACA57830). Alignments were performed with the pairwiseAlignment method from the R BioStrings package in local alignment mode (Smith-Waterman) using the BLOSUM80 amino acid substitution matrix. The sequence motif was generated using the ggseqlogo package in R[26].

**Preparation of the RABV G ectodomain for BLI.** The ectodomain construct comprising residues K1-S403 (strain 9147FRA, GenBank: AF401286) with a C-terminal Strep tag (sequence: GGWSHPQFEK) and its mutants D190S, K226T, V230M and V230K were prepared in the same way as domain III for

crystallization. Primers for cloning and mutagenesis are listed in Supplementary Table 1. In order to improve secretion into the culture supernatant of stably transfected S2 cells, point mutations F74H and W121H were introduced into the fusion loops and an untagged scFv of RVC58, which binds to antigenic site III[9], was co-expressed. During purification, the RVC58 scFv was removed again by gel permeation chromatography in 20 mM MES pH 6.5, 150 mM NaCl, 10% v/v glycerol—a condition, which was found to destabilize the interaction sufficiently for separation. The final sample was diluted to a protein concentration of 0.5 mg mL$^{-1}$ in 50 mM Tris-Cl pH 8.0, 150 mM NaCl, 10% v/v glycerol prior to analysis.

**Biolayer interferometry.** Biolayer interferometry (BLI) was carried out on an Octet RED384 instrument (ForteBio). Recombinant RVC20 wild-type or variants at a concentration of 10 μg mL$^{-1}$ were immobilized on anti-human IgG Fc Capture (AHC) biosensors (ForteBio). Running buffers were either 50 mM Tris-Cl pH 8.0, 50 mM MES pH 7.0 or 50 mM MES pH 5.5, each in 150 mM NaCl, 10% (v/v) glycerol and 0.2 mg mL$^{-1}$ bovine serum albumin. The loaded and equilibrated biosensors were transferred into solutions containing different antigen concentrations ranging from 3.7 nM to 300 nM in the indicated buffers. Association and dissociation were monitored for 20 min each, and the data were recorded in triplicates. Nonspecific binding and baseline drift were accounted for by subtracting the readings of reference runs omitting the mAb immobilization step and of reference sensors dipped into blank buffer during the association step, respectively. Dissociation was exceedingly slow for most samples, thus no reliable $K_D$ values could be obtained from kinetic analysis. Instead, apparent steady-state $K_D$ values were approximated from fitting the response levels after 20 min of association using the pro Fit software (QuantumSoft).

**Recombinant IgG production**. UA sequences were determined with reference to the IMGT database[27] and produced by gene synthesis (Genscript). VH and VK sequences of RVC20 antibody and derived variants were cloned into human IgG1 and IgK expression vectors and recombinant mAbs were produced by transient transfection of ExpiCHO cells (Thermo Fisher Scientific, Cat# A29127), purified by Protein A chromatography (GE Healthcare) and desalted against PBS.

**Reverse genetics and virus titration**. Tha-GFP recombinant virus[28] is based on the wild isolate 8743THA, EVAg collection, Ref-SKU: 014V-02106, isolated from a human bitten by a dog in 1983 in Thailand. Mutations D190S, K226T, V230M and V230K were introduced into the G gene using the Phusion Site-Directed Mutagenesis Kit (Thermo Scientific) according to the manufacturer's instructions. All mutagenesis primers are listed in Supplementary Table 2. Recombinant RABV wild-type or mutants were rescued[19] by transfection of BSR-T7 cells with the complete genome (2.5 µg) together with plasmids N-pTIT (2.5 µg), P-pTIT (1.25 µg) and L-pTIT (1.25 µg)[29]. At 6 days post-transfection, the cells were serially passaged every 3 days. When 100% of the cells were infected, the supernatant was harvested and titrated on BSR cells. The infection was monitored by immunofluorescence using a FITC-conjugated anti-RABV nucleocapsid antibody (Bio-Rad, Cat# 3572112) according to the manufacturer's instructions.

Titrations were performed on BSR cells by the fluorescent focus method[30]. A total of 20 µL of serial dilutions (1 to 5) of virus were inoculated in duplicates on $5 \times 10^4$ BSR cells and incubated at 37 °C. At 40 h post-infection, the medium was removed, the cells were fixed with 80% acetone and incubated with the FITC-conjugated anti-RABV nucleocapsid antibody (Bio-Rad, Cat# 3572112) according to the manufacturer's instructions. The number of fluorescent foci was determined under a fluorescent microscope and the titer was calculated in fluorescent focus units per milliliter (FFU mL$^{-1}$). To determine growth curves for the recombinant viruses, BSR cells were inoculated with each virus at MOI = 0.1 and supernatants were recovered at 24 h, 48 h and 72 h post-infection for titration on BSR cells.

**Virus neutralization test**. A total of $2 \times 10^3$ FFU of RABV wild-type or mutants were incubated with different concentrations of RVC20 (mature or variants) in DMEM with 10% fetal bovine serum for 1 h at 37 °C in 96-well plates (Greiner Bio-One, #655090); $1 \times 10^4$ BSR cells were then added to each well and the plates were incubated at 37 °C (final MOI = 0.2). After 48 h, the cells were fixed with 4% PFA, washed in PBS and the nuclei were counterstained with 20 µM Hoechst 33342. Image acquisitions of 16 fields per well (totaling 26.6 mm$^2$ per well) were performed on the automated confocal microscope Opera Phenix (Perkin Elmer) using the 10× objective. The data were transferred to the Columbus Image Data Storage and Analysis System (Perkin Elmer) and the percentage of GFP-positive cells was determined. IC$_{50}$ values were determined by nonlinear regression analysis (GraphPad Prism) from three independent experiments.

**Pseudotype neutralization test**. Lentiviral pseudotypes[9] were produced in HEK293T clone 17 cells. Neutralization assays were undertaken on BHK-21 clone 13 cells. In a 96-well white plate, pseudotyped virus that resulted in an output of $30–70 \times 10^3$ relative light units (RLU) was incubated with dilutions of RVC20 for 1 h at 37 and 5% CO$_2$ before the addition of 10,000 BHK-21 cells. These were incubated for additional 72 h, after which supernatant was removed and SteadyGlo reagent (Perkin Elmer) was added. Luciferase activity was detected 5 min later by reading the plates on a Synergy H1 microplate luminometer (BioTek). The reduction of infectivity was determined by comparing the RLU in the presence and absence of antibodies and expressed as percentage of neutralization. IC$_{50}$ of neutralization of antibodies was calculated by non-linear regression fitting using GraphPad Prism 6 software (GraphPad Software Inc., San Diego. CA). Mutants of Lagos bat virus (strain LBV.NIG56-RV1, GenBank: EF547431) G genes were generated by gene synthesis and confirmed by sequence analysis. The resulting G genes were subsequently used to generate pseudotyped viruses and titrated on BHK-21 cells to ensure the mutations did not affect the binding and entry function of the G proteins.

**Flow cytometry**. The coding sequence of RABV G (isolate 8743THA, EVAg collection, Ref-SKU: 014V-02106) was amplified from cells infected with Tha-GFP recombinant virus and was cloned into the phCMV vector (GenBank: AJ318514) using In-Fusion cloning (Clontech) to generate phCMV-8743. The cloning primers are listed in Supplementary Table 2. A total of $2 \times 10^6$ HEK293-T cells were transfected in suspension with 100 ng of phCMV-8743 mixed with 1900 ng of pQCXIP-empty using Lipofectamine 2000 (Thermo Fisher); 18 h after transfection, the cells were lifted using PBS + 0.1% EDTA. The cells were subsequently resuspended in 60 mM MES + 100 mM NaCl at pH 7.0 or pH 5.5, and were equilibrated for 15 min at 37 °C, shaking at 1100 rpm to avoid cell–cell fusion. Equilibration was followed by a mAb association step at 800 ng mL$^{-1}$ of isotype control mGO53, RVC20 or RVC68 in the indicated buffers for 15 min at room temperature with shaking. The cells were next washed in the indicated buffers and incubated for an additional 15 min shaking in the same buffers. After two washing steps in PBS, the cells were stained with secondary Goat anti-Human IgG, Alexa Fluor 568 (Invitrogen, #A21090) diluted 1:500 in PBS + 1% bovine serum albumin. Cells were

washed and fixed for 10 min in 4% PFA prior to fluorescence measurement on an Attune NxT Flow Cytometer (Thermo Fisher). The gating strategy is exemplified in Supplementary Fig. 5.

**GFP-Split fusion assay**. Cell–cell fusion experiments were performed using the HEK 293T GFP-split system[31]. GFP$_{1-10}$ and GFP$_{11}$-expressing HEK 293T cells were mixed at a 1:1 ratio. A total of $6 \times 10^4$ cells per well were suspended in 96-well plates (µClear, #655090) and were transfected with 5 ng of phCMV-8743 expressing RABV G mixed with 95 ng of DNA with pQCXIP-empty using Lipofectamine 2000 (Thermo Fisher). MAbs were added to the medium at the indicated concentrations on the day of transfection; 18 h after transfection, 2/3 of the medium was exchanged for 60 mM MES + 100 mM NaCl at several pH values (7.0, 6.0, 5.5 or 5.0). Following incubation for 15 min at 37 °C, the medium was again exchanged for DMEM 10% FBS at pH 7, and the cells were left to fuse for an additional 90 min. Thirteen images per well corresponding to 90% of the well surface were acquired on an Opera Phenix High-Content Screening System (PerkinElmer) and the surface area covered with GFP-positive cells was determined.

**Statistical analysis**. Statistical analysis was performed in GraphPad Prism 6 using one-way ANOVA followed by Tukey's multiple comparisons test with $\alpha = 0.05$, except for the data presented in Fig. 3e, where two-way ANOVA was applied.

**Reporting summary**. Further information on research design is available in the Nature Research Reporting Summary linked to this article.

## Data availability
The authors declare that the data supporting the findings of this study are available within the paper and the Supplementary Information. The crystal structure of the RVC20 scFv/RABV G domain III complex from this study is available in the PDB with the accession code: 6TOU (https://doi.org/10.2210/pdb6TOU/pdb). The source data underlying Figs. 1e, 1f, 2c, 3d, 3e and Supplementary Fig. 3 are provided as a Source Data file.

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

## Acknowledgements

This work was funded by Institut Pasteur. F.A.R. received additional funding from the CNRS, the Labex IBEID (ANR-10-IHUB-0002), the GIS IBiSA (Infrastructures en biologie santé et agronomie), grant ANR-13-ISV8-0002-01 and the Région Ile de France in order to support the work of J.H. (Domaine d'intérêt majeur innovative technologies for life sciences, DIM 1HEALTH). O.S. received funding from ANRS, Sidaction, the Vaccine Research Institute (ANR-10-LABX-77), the Labex IBEID (ANR-10-IHUB-0002), the "TIMTAMDEN" ANR-14-CE14-0029, the "CHIKV-Viro-Immuno" ANR-14-CE14-0015-01, L'Oréal Sponsorship, the CNRS and the Gilead HIV cure program. H.B., F.L. and G.D.M. received funding from the INFECT-ERA 2016 project ANR 16-IFEC-0006-01 ToRRENT. We thank Yves Gaudin and Eduard Baquero for reagents and advice on the RABV G fusion assay, the staff of the crystallography platform at Institut Pasteur for robot-driven crystallization screening, the staff of synchrotron beamline PX2 at SOLEIL (St. Aubin, France) for help during data collection, and Lauriane Kergoat and Margot Dropy for technical assistance. Part of this work was performed at the UtechS Photonic BioImaging (PBI) platform, member of France Life Imaging network (grant ANR-11-INBS-0006).

## Author contributions

J.H., J.B., O.S., D.C., H.B. and F.A.R. designed the experiments. J.H. designed and prepared the recombinant antigens and the scFv, and performed the biophysical and crystallographic analysis, to which A.H. and P.E contributed. J.B. performed the flow cytometry and the fusion experiment. F.L. and G.D.M. performed the reverse genetics and the RABV neutralization assays. A.M. prepared the LBV pseudotyped viruses, performed the respective neutralization assays, and prepared the mAbs. A.T. and L.S. analyzed the RABV epitope conservation. J.H. and F.A.R. wrote the manuscript with contributions by all authors.

## Competing interests

A.M., L.S., A.T. and D.C. are employees of Vir Biotechnology Inc. and may hold shares in Vir Biotechnology Inc. H.B. and D.C. are authors of two patents that concern Mab RVC20 (PCT /EP2015/002305: Antibodies that potently neutralize rabies virus and other lyssaviruses and uses thereof; PCT/EP2019/078439: Antibodies and methods for treatment of lyssavirus infection). The other authors declare no competing interests.
