## [Peer Review File · Nature Communications]

Reviewers' Comments:

Reviewer #1:

Remarks to the Author:

Hellert et al. presented a structural study of a broadly neutralizing antibody RVC20 for rabies viruses (RABV). The discovery and functional characterization of RVC20, including neutralizing activity and epitope mapping, were previously reported by some of the authors of the present study (De Benedictis et al. 2016). In this study, the structure of RVC20 in complex with domain III of RABV surface glycoprotein (G) was determined to better than 2.7 Å. As shown in the previous study (De Benedictis et al. 2016), RVC20 can neutralize some non-RABV lyssaviruses, but not certain distantly related lyssaviruses such as Lagos bat virus. With the availability of the structural information, the authors in this study identified and validated two residues on the G protein of Lagos bat virus that prevent it from being neutralized by RVC20. The authors also identified several somatic mutations on RVC20 that are important for affinity maturation. Structural analysis suggested that RVC20 binds to RABV G protein in its prefusion state, but not its post-fusion state, which was validated by biolayer interferometry and flow cytometry experiments. Finally, the authors showed that RVC20 inhibited the conformational rearrangement of RABV G protein, which in turn blocked its membrane fusion function.

Overall, this study presents solid data and the paper is well written. Prior to this study, no structural information was available for the RABV surface glycoprotein (G). Therefore, the structural data in this study are important for the field, albeit that only a domain of RABV has been determined structurally. In addition, this study provides important information for antiviral design against RABV, as well as other lyssaviruses.

In Table S1, please make the following changes:

α , β , γ (°)

Please replace "90.00, 90.00, 90.00" with "90, 90, 90".

Please add Rpim and number of unique reflections measured (in addition to the number used for refinement).

Please add Wilson B and Ramachandran statistics.

Supplementary Table 2:

Please truncate the BSA values to integers or to at most one decimal point.

Reviewer #2:

Remarks to the Author:

The authors report the X-ray crystal structure of the ectodomain of the Rabies virus G protein bound to the scFv of a neutralizing antibody, RVC20. The structure reveals that the antibody binds to a conserved surface in G. The authors show that the antibody, through its binding mode, likely prevents conformational changes in G that are important for it to mediate membrane fusion. The work is well-executed, the text is well written, the data and figures are of high quality, and the work is important and of broad interest. I have only minor comments/suggestions.

Minor comments:

- The authors could provide a figure that shows sequence variation as a function of color intensity on G (e.g., mapping sequence conservation onto a surface representation of G) based on the sequences they used in their phylogenetic analysis. This diagram could help address the question, is the part of DIII bound in the structure the most conserved surface of G?

- The fog effect in Fig. 3b is so pronounced that it obscures too much of the ribbon diagram. This effect should be decreased for improved clarity.

- Line 164 – “The good conservation of the RVC20 epitope...” The term “good” seems odd here. Do the authors mean, instead, “The high degree of conservation...”?

- Line 173 of the text, where it is written that “In the spirit of recently introduced drug development strategy against influenza virus hemagglutinin...” and where citations are provided, the text should be made more clear to specify the details of the approach. The reviewer had to pull up the full reference to understand that the authors were perhaps referring to a high throughput screen to identify small molecules that target protein-protein interfaces. The authors should, instead, state their proposed approach.

REVIEWERS' COMMENTS:

Reviewer #1 (Remarks to the Author):

Hellert et al. presented a structural study of a broadly neutralizing antibody RVC20 for rabies viruses (RABV). The discovery and functional characterization of RVC20, including neutralizing activity and epitope mapping, were previously reported by some of the authors of the present study (De Benedictis et al. 2016). In this study, the structure of RVC20 in complex with domain III of RABV surface glycoprotein (G) was determined to better than 2.7 Å. As shown in the previous study (De Benedictis et al. 2016), RVC20 can neutralize some non-RABV lyssaviruses, but not certain distantly related lyssaviruses such as Lagos bat virus. With the availability of the structural information, the authors in this study identified and validated two residues on the G protein of Lagos bat virus that prevent it from being neutralized by RVC20. The authors also identified several somatic mutations on RVC20 that are important for affinity maturation. Structural analysis suggested that RVC20 binds to RABV G protein in its prefusion state, but not its post-fusion state, which was validated by biolayer interferometry and flow cytometry experiments. Finally, the authors showed that RVC20 inhibited the conformational rearrangement of RABV G protein, which in turn blocked its membrane fusion function.

Overall, this study presents solid data and the paper is well written. Prior to this study, no structural information was available for the RABV surface glycoprotein (G). Therefore, the structural data in this study are important for the field, albeit that only a domain of RABV has been determined structurally. In addition, this study provides important information for antiviral design against RABV, as well as other lyssaviruses.

In Table S1, please make the following changes:

α , β , γ (°)

Please replace “90.00, 90.00, 90.00” with “90, 90, 90”.

- The change has been incorporated.

Please add R_{pim} and number of unique reflections measured (in addition to the number used for refinement).

- R_{pim} and the number of unique reflections measured have been added to the table.

Please add Wilson B and Ramachandran statistics.

- We agree with the reviewer that this information deserves a place in the statistics table, but in order to comply with the guidelines for X-ray statistics tables in *nature* journals, we provide the Wilson *B* and Ramachandran statistics in the respective subsection of the Methods.

Supplementary Table 2:

Please truncate the BSA values to integers or to at most one decimal point.

- We truncated the BSA values to integers and the conservation percentages to one decimal point.

Reviewer #2 (Remarks to the Author):

The authors report the X-ray crystal structure of the ectodomain of the Rabies virus G protein bound to the scFv of a neutralizing antibody, RVC20. The structure reveals that the antibody binds to a conserved surface in G. The authors show that the antibody, through its binding mode, likely prevents conformational changes in G that are important for it to mediate membrane fusion. The work is well-executed, the text is well written, the data and figures are of high quality, and the work is important and of broad interest. I have only minor comments/suggestions.

Minor comments:

- The authors could provide a figure that shows sequence variation as a function of color intensity on G (e.g., mapping sequence conservation onto a surface representation of G) based on the sequences they used in their phylogenetic analysis. This diagram could help address the question, is the part of DIII bound in the structure the most conserved surface of G?

- We agree that such a figure would be interesting, but as long as no structure of a more complete RABV G ectodomain fragment is available yet, we cannot provide an accurate representation. Homology models based on the more complete VSV G structure will not be reliable enough for this purpose.

- The fog effect in Fig. 3b is so pronounced that it obscures too much of the ribbon diagram. This effect should be decreased for improved clarity.

- Fig. 3b now shows less fog on the surface of the structure.

- Line 164 – “The good conservation of the RVC20 epitope...” The term “good” seems odd here. Do the authors mean, instead, “The high degree of conservation...”?

- We thank the reviewer for having spotted this, and changed the text according to his proposition.

- Line 173 of the text, where it is written that “In the spirit of recently introduced drug development strategy against influenza virus hemagglutinin...” and where citations are provided, the text should be made more clear to specify the details of the approach. The reviewer had to pull up the full reference to understand that the authors were perhaps referring to a high throughput screen to identify small molecules that target protein-protein interfaces. The authors should, instead, state their proposed approach.

- We now rephrased this section to be more explicit.